# Physicochemical Properties and Biological Activity of Active Films Based on Corn Peptide Incorporated Carboxymethyl Chitosan

**Liyan Wang** [1] **, Liang Lei** [1]**, Kang Wan** [1]**, Yuan Fu** [1] **and Hewen Hu** [1,2,*]

[1] College of Food Science and Engineering, Jilin Agricultural University, 2888 Xincheng Street, Changchun 130118, China; wangliyan@jlau.edu.cn (L.W.); LEILIANG15643499436@jlau.edu.cn (L.L.); wankang@jlau.edu.cn (K.W.); 20201617@mails.jlau.edu.cn (Y.F.)

[2] College of Humanities (College of Home Economics), Jilin Agricultural University, 2888 Xincheng Street, Changchun 130118, China

[*] Correspondence: huhewen@jlau.edu.cn; Tel.: +86-135-1449-2805

**Abstract:** Active films based on carboxymethyl chitosan incorporated corn peptide were developed, and the effect of the concentration of corn peptide on films was evaluated. Physicochemical properties of the films, including thickness, opacity, moisture content, color, mechanical properties, water vapor permeability, and oil resistance, were measured. Biological activities of the films, including the antioxidant and antibacterial activities, were characterized in terms of 2, 2-diphenyl-1-picrylhydrazyl free radical scavenging activity, reducing power, the total antioxidant activity, and the filter disc inhibition zone method. The results indicated that the incorporation of corn peptide caused interactions between carboxymethyl chitosan and corn peptide in Maillard reaction and gave rise to the films light yellow appearance. Compared with the Control, the degree of glycosylation, browning intensity, thickness, opacity, tensile strength, antioxidant activity, and antibacterial activity of films were increased, but the elongation, vapor permeability, and oil resistance of films were decreased. The films based on corn peptide and carboxymethyl chitosan can potentially be applied to food packaging.

**Keywords:** carboxymethyl chitosan; corn peptide; antioxidant activity; antibacterial activity; film

## 1. Introduction

Packaging plays an important role in maintaining food quality and safety [1]. However, non-biodegradable packaging has caused serious environmental problems. The world's oceans and arable land are suffering from plastic pollution, which puts tremendous pressure on the ecosystem [2]. Therefore, lots of researchers are committed to developing biodegradable and environmentally friendly packaging materials [3,4], such as chitosan and its derivatives, pectin, starch, and cellulose derivatives [5–7].

Carboxymethyl chitosan (CMCS) is obtained by importing carboxymethyl groups into the amino, primary, and secondary hydroxyl sites of the glucosamine units, which is an amphoteric derivative of chitosan [8]. CMCS possesses various features, such as biocompatible, non-toxic, antibacterial, film-forming, and degradable [9,10], and widely used in the field of food packaging [11,12]. It has been proved that emulsion based on CMCS and natamycin could inhibit *Alternata alternara* and prolong the shelf life of jujube fruits [13]. The addition of CMCS could improve the mechanical properties and thermal stability of the rice starch film [14]. However, pure CMCS film is fragile and hydrophilic, and its antioxidant and antibacterial activity is insufficient [15]. Fortunately, the -OH, -NH$_2$, and -COOH groups on the CMCS backbone is beneficial to the chemical modification to improve the antioxidant and antibacterial activity [16,17].

Corn peptide (CP) is a kind of corn protein hydrolysate. CP has some incomparable physiological function, such as antioxidant activities [18], antihypertensive effect [19],

and metabolizing alcohol [20]. It has been reported that the mixture of corn peptide and wheat peptide would be used to prevent diabetes in non-obese diabetic (NOD) mice [21]. However, the use of CP modified CMCS has not been investigated.

Maillard reaction is a chemical reaction between the carbonyl group of reducing sugars and amino acids, peptides, or proteins [22], and its products possess significant antioxidant and antibacterial activities [23–25]. For example, the combination of chitosan and soy protein enhanced antioxidant and antibacterial activity [26]. Nevertheless, using Maillard reaction products to develop films based on CMCS and CP has not been reported.

Therefore, the purpose of this study was to evaluate the effects of CP concentration on physicochemical properties (thickness, opacity, moisture content, color, mechanical properties, water vapor permeability, and oil resistance) and biological activities (antioxidant and antibacterial activity) of films based on carboxymethyl chitosan and corn peptide.

## 2. Materials and Methods

### 2.1. Materials

Carboxymethyl chitosan (CMCS) with a degree of carboxylation of 80% (commercial code: MB1905, 10 mPa.s~80 mPa.s) was provided from Dalian Meilun Biotechnology Co., Ltd., China (Dalian, China). Corn peptide was purchased from Wuxi Shengshi Hongcheng Biotechnology Co., Ltd., (Wuxi, China). The corn peptide used in this study is light yellow powder with a content of 99%. Calcium chloride and potassium nitrate were purchased from Beijing Beihua Co., Ltd., (Beijing, China). 2, 2-diphenyl-1-picrylhydrazyl (DPPH) was purchased from Sigma-Aldrich Company (St. Louis, MI, USA).

### 2.2. Preparation of Maillard Reaction Products

The CMCS solution (2 wt%) was prepared by dissolving CMCS into distilled water with stirring of 5 kr/min for 30 min at 4 °C. The CP solutions at concentration of 0%, 10%, 20%, 30%, and 40% of the CMCS were prepared by mixing CP with distilled water for 30 min at 4 °C. The mixture solutions were prepared by mixing the CMCS solution and the CP solutions at weight ratio of 1:1. The mixtures were pre-freezed at −20 °C for 12 h, then freeze-dried at −50 °C for 24 h in Freeze dryer (Beijing Bo Yikang Experimental Instrument Co., Ltd., (Beijing, China)), then kept at 60 °C and 79% relative humidity (RH) for 48 h in constant temperature and humidity box (Shanghai Zhicheng Analytical Instrument Manufacturing Co., Ltd., (Shanghai, China)). After freeze-dried at −50 °C for 8 h, the Maillard reaction products was obtained and then stored at 4 °C.

### 2.3. Preparation of Films

The different components Maillard reaction products (1 g) were dissolved in deionized water (30 g), mixed thoroughly, stirred with 800 rpm for 60 min at 25 °C, added the glycerol (0.25 g) as a plasticizer, and degassed in Ultrasonic cleaner (Shanghai Experimental Instrument Factory Co., Ltd., Shanghai, China). It was cast into the petri dish (diameter 90 mm) and dried at 25 °C and 30% RH for 48 h. These peeled films were placed at room temperature 25 °C and 75% relative humidity for 48 h in constant temperature and humidity box before measured. The obtained films were named according to the concentration of CP as follows: Control, CP-10, CP-20, CP-30, and CP-40, respectively.

### 2.4. Characterization

#### 2.4.1. Measurement of the Browning Intensity of the Films

The browning intensity was measured according to the method of Vhangani et al. [27].

#### 2.4.2. Measurement of the Degree of Glycosylation of the Films

The degree of glycosylation of films was measured by the o-phthalic aldehyde (OPA) method with certain modifications [28]. OPA (40.0 mg) was dissolved in methanol (1.0 mL), and then added 2.5 mL of sodium dodecyl sulfate (20% *w/v*), 25.0 mL of borax (0.01 mol/L), 100 μL of β-mercaptoethanol, then diluted the volume to 50 mL with deionized water.

4.0 mL of the OPA reagent was taken in the test tube, 200 μL of solution containing the film samples (2.0 mg/mL) was added and mixed thoroughly, and it was placed in 35 °C water bath in dark for 2 min, then measured the absorbance at 340 nm. Another 4.0 mL of OPA reagent was mixture with 200 μL of deionized water as the CMCS. The same method used lysine instead of the sample to make a standard curve. The degree of glycosylation of the films was calculated by the following formula:

$$\text{Degree of glycosylation (\%)} = \frac{C_0 - C_1}{C_0} \times 100\% \tag{1}$$

where $C_0$ and $C_1$ (mol/L) the content of free amino groups in the solution before and after Maillard reaction, respectively.

### 2.4.3. Thickness, Opacity, and Moisture Content

The thicknesses of five random points of the films were measured, and the average value was expressed as the thickness of films.

The films were cut into long strips (1 cm × 4 cm) and placed in the test cell of the spectrophotometer. An empty test cell was used as a reference. Opacity of the films was determined by using an UV–Visible spectrophotometer (Lambda 365) at 600 nm [29]. The opacity of the films was calculated by the following formula:

$$O = \frac{\text{Abs}_{600}}{d} \tag{2}$$

where O is the opacity, $\text{Abs}_{600}$ is the value of absorbance at 600 nm, and d is the films thickness (mm).

The moisture content of the films was measured by Oven drying method (105 °C for 10 h). The moisture content was calculated by the following formula:

$$\text{Moisture content (\%)} = \frac{M_w - M_d}{M_w} \times 100\% \tag{3}$$

where $M_w$ is the weight of the films in 75% RH and $M_d$ is dry weight of the films.

### 2.4.4. Color Properties

HunterLab ColorFlex (Shanghai Xinlian Creative Electronics Co., Ltd., China) was used in order to assess the color changes in films. A white standard color plate (L* = 94.50, a* = −0.84, b* = 0.65) was used as background. L* represents lightness or blackness, a* represents redness or greenness, and b* represents yellowness or blueness values. The total color difference (ΔE*) was calculated by the following formula:

$$\Delta E = \sqrt{\Delta a^2 + \Delta b^2 + \Delta L^2} \tag{4}$$

where ΔL = L*standard − L*sample, Δa = a*standard − a*sample, Δb = b*standard − b*sample.

### 2.4.5. Mechanical Properties

Mechanical properties of the films were measured using Electronic universal testing machine (Shanghai Qingji Instrument Technology Co., Ltd., (Shanghai, China)). These films were cut into a dumbbell shape with a vise and a knife, the narrow parallel part was 0.4 cm wide and the total length was 5.0 cm. Each film was tested in 3 parallels, and the results were averaged.

### 2.4.6. Water Vapor Permeability (WVP)

The WVP of films was measured according to the method of Wang et al. [30]. It was based on a special aluminum cup with a depth of 1.3 cm and an inner diameter of

6.0 cm (exposed area: 28.26 cm$^2$). The cup was filled with anhydrous calcium chloride (2.0 g) to provide 0% RH (R$_1$), sealed with films, and then placed in constant temperature and humidity box to maintain 95% RH (R$_2$). The WVP of films was calculated by the following formula:

$$WVP = \frac{slope \times d}{A \times p \times (R_1 - R_2)} \quad (5)$$

where WVP is the water vapor permeability (g mm m$^{-2}$ day$^{-1}$ kPa$^{-1}$), slope is linear portion of the weight gained versus time plot (g/h), d is the films thickness (mm), A is the exposed area (m$^2$), p is the saturation vapor pressure of water at the test temperature (25 °C), R$_1$ is the relative humidity in constant temperature and humidity box, R$_2$ is the relative humidity inside the cup.

### 2.4.7. Oil Resistance

The oil resistance of the films was measured according to the method of Wang et al. [30]. The filter paper (diameter 6.0 cm) was placed in an oven at 50 °C for 6 h until constant weight. The test tube (5.0 mL soybean oil) was sealed by the film, fixed with a string, and inverted down on the filter paper into the drying tank. The oil absorption rate calculated by the following formula:

$$OAR\ (\%) = \frac{M - M_0}{M_0} \times 100\% \quad (6)$$

where OAR (%) is the oil absorption rate, M$_0$ and M are the filter paper weight before and after oil absorption, respectively.

### 2.4.8. Antioxidant Activity
#### DPPH Free Radical Scavenging Activity

The DPPH free radical scavenging activity was measured according to the method of Wang et al. [30]. DPPH assay solution was developed by mixing 9.0 mL of films extract with 3.0 mL of methanol solution of DPPH (10$^{-3}$ mol/L). After shaken in an oscillator (Ronghua Instrument Manufacturing Co. Ltd., (Jintan, China)) for 1 min, the mixture was incubated at room temperature for 30 min in the dark. The absorbance of the DPPH assay solution was measured by the UV spectrophotometer at 517 mm. Each sample was tested three times and the average value was taken. The DPPH scavenging activity was calculated by the following formula:

$$DPPH\ scavenging\ activity = \frac{A_{DPPH} - A_S}{A_{DPPH}} \times 100 \quad (7)$$

where A$_{DPPH}$ is the absorbance value of the DPPH methanol solution and A$_S$ is the absorbance value of the DPPH assay solution.

#### Reducing Power

The ability of the films to reduce iron (III) was measured according to the method of Kchaou et al. [31], although slightly modified. The films (m = 0.1 g) were immersed in 5 mL of distilled water, 12.5 mL of 0.2 M phosphate buffer (pH 6.6), and 12.5 mL of 1% (*w/v*) potassium ferricyanide. After incubating at 50 °C for 3 h in the dark, 5 mL of 10% (*w/v*) trichloroacetic acid was added to the mixture. Then, 12.5 mL of the supernatant was mixed with 15 mL reagent solution (distilled water and ferric chloride (0.1% *w/v*); 5:1). After a reaction time of 10 min, the absorbance of the resulting solution was measured at 700 nm. The higher absorbance of the reaction mixture presented higher reducing power.

#### Total Antioxidant Activity

The total antioxidant activity of the films was measured according to the method of Sa et al. [32], although slightly modified. The films were cut into long strips (m = 0.1 g),

immersed in 1 mL of deionized water, and homogenized with 10 mL of reagent solution (0.6 M sulphuric acid, 28 mM sodium phosphate, and 4 mM ammonium molybdate). After incubating for 90 min at 90 °C, the absorbance values were measured at 695 nm. 1 mL of the deionized water was used as a control.

### 2.4.9. Antibacterial Activities

Antibacterial activities of the films were evaluated against Gram-negative bacteria, Escherichia coli (*E. coli*), and Gram-positive bacteria, Staphylococcus aureus (*S. aureus*), using the method of the filter disc inhibition zone. The sterile filter paper was cut into discs (diameter 6 mm) and immersed in the films extract (200 μL) for 1 h. The tested strain was cultured and activated in a shaker at 37 °C and 80 r/min for 14 h, diluted 100 times with sterile saline. A total of 1.0 mL of the bacterial solution was mixed with the liquid nutrient agar to solidify before poured it into the petri dish. The filter paper soaked in the films extract was gently placed on the solidified petri dish with tweezers, and then put it in the refrigerator at 4 °C for 30 min. Finally, the petri dish was incubated at 37 °C for 24 h before we measured the diameter of the inhibition zone. The experiment was repeated in triplicate for each sample.

### 2.4.10. Attenuated Total Reflectance-Fourier Transform Infrared Analysis (ATR-FTIR)

A Nexus 670 FTIR spectrometer (Nicolet, Waltham, MA, USA) with ATR accessory was used to measure the ATR-FTIR spectrum of the obtained films to study the interaction of CMCS and CP. A spectral resolution of 4 cm$^{-1}$ was used, and 64 scans were performed for each spectrum in the range of 4000 to 400 cm$^{-1}$.

### *2.5. Statistical Analysis*

The SPSS Statistics analyzed the comparative mean, and the differences between factors and levels were evaluated by one-way analysis of variance (ANOVA). The Duncan's New Multiple Range Test was used to compare the significant differences between the means ($p < 0.05$).

## 3. Results and Discussion

### *3.1. Degree of Glycosylation and Browning Intensity of the Films*

Degree of glycosylation and browning intensity are important indicators to judge Maillard reaction. Figure 1 showed the degree of glycosylation of the films determined from regression equation of the calibration curve (y = 0.0032x + 0.0129; r$^2$ = 0.990) and the change of the degree of glycosylation with the concentration of CP. The degree of glycosylation of Control was $1.02 \pm 0.75\%$. This is probably because amino and carboxyl groups in Control occurred in a Maillard reaction. With the increase of corn peptide concentration, the degree of glycosylation of the films increased but then decreased. The highest degree of glycosylation ($26.92 \pm 0.14\%$) was found in CP-30, this might be the reaction of protein and polysaccharides which resulted in a decrease in free amino group content, thereby increasing the degree of glycosylation [33]. The degree of glycosylation of CP-40 was lower (31.14%) than CP-30. The possible reason was that the Maillard reaction products increased the viscosity of the system, limiting the molecular motion of amino groups and carboxyl groups [34].

The extent of Maillard reaction can be monitored by browning intensity. Figure 1 showed the change of the browning intensity with the concentration of CP. It can be seen that, the browning intensity increased significantly ($p < 0.05$) with the increase of CP concentration due to the production of melanoidins [27].

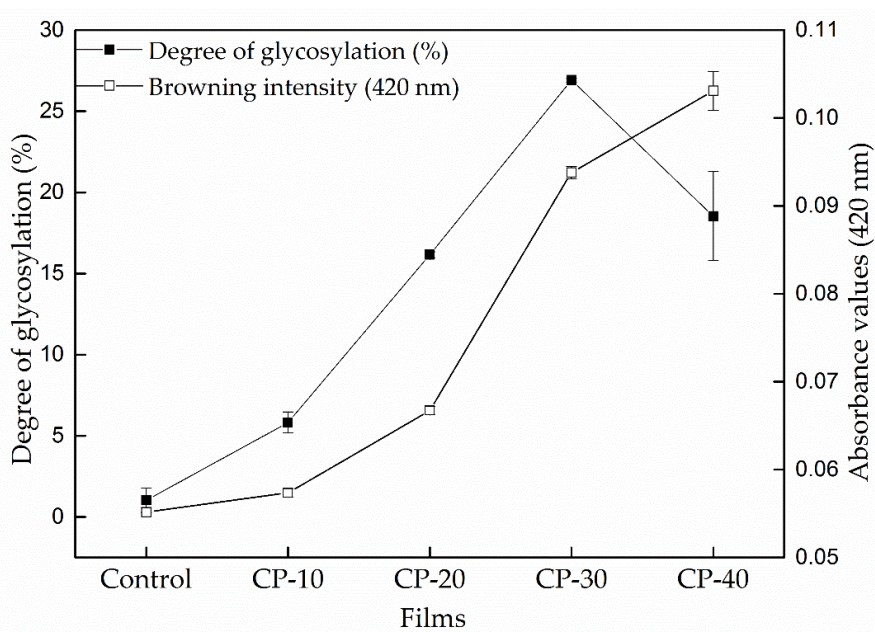

**Figure 1.** Degree of glycosylation and browning intensity of films.

*3.2. Thickness, Opacity, and Moisture Content*

The thickness, moisture content, and opacity of the films are shown in Table 1. With the increase of CP concentration, the thickness of films increased significantly ($p < 0.05$). The thickness of films could be related with the type of biopolymer selected and the amount of CP incorporated [15].

**Table 1.** Thickness, opacity, and moisture content of films.

| Films | Thickness (mm) | Opacity (A mm$^{-1}$) | Moisture Content (%) |
|---|---|---|---|
| Control | 0.09 ± 0.14 [a] | 0.59 ± 0.49 [a] | 22.62 ± 0.69 [a] |
| CP-10 | 0.13 ± 0.10 [b] | 0.68 ± 0.30 [a] | 20.34 ± 0.35 [b] |
| CP-20 | 0.15 ± 0.54 [b] | 1.03 ± 0.72 [b] | 18.43 ± 0.45 [c] |
| CP-30 | 0.15 ± 0.22 [b] | 1.25 ± 0.36 [c] | 17.33 ± 0.18 [c,d] |
| CP-40 | 0.16 ± 0.26 [b] | 1.36 ± 0.44 [c] | 18.77 ± 0.30 [c] |

Values are given as mean ± standard deviation. Different letters in the same column indicate significantly different ($p < 0.05$) when analyzed by Duncan's New Multiple Range Test.

With the increase of CP concentration, the opacity of the films increased significantly ($p < 0.05$), this was expected since the change in opacity is attributed to the CP concentration. Similar phenomena were discovered in carboxymethyl chitosan–quercetin composite films [15]. The higher opacity values, the lower transparency. Control (without CP) was more transparent than those containing CP.

After incorporation of CP, the moisture content of the films was significantly ($p < 0.05$) decreased. The combinations of proteins and polysaccharides would release water molecules in Maillard reaction, thereby decreasing the moisture content of the films [25]. CP-30 possesses the lowest moisture content, which is 23.39% lower than that of Control. However, there were no significant difference ($p > 0.05$) among CP-20, CP-30, and CP-40.

*3.3. Color Properties*

Color properties were significant to the appearance of the films. The L*, a*, and b* values of the films were listed in Table 2. With the increase of CP concentration, the value of L* and a* decreased significantly ($p < 0.05$), while the value of b* increased, and the color of films transform visually from colorless and transparent to light yellow, which was related to the interaction between CP and CMCS through Maillard reaction [31]. The total

color difference (ΔE*) of films were significantly ($p < 0.05$) increased, the higher ΔE* gave rise to more colored films.

**Table 2.** Color properties of the films.

| Films | L* | a* | b* | ΔE* |
|---|---|---|---|---|
| Control | 90.26 ± 0.29 [d] | −1.71 ± 0.05 [d] | 7.31 ± 0.51 [a] | 7.75 ± 0.57 [a] |
| CP-10 | 89.85 ± 0.22 [c,d] | −1.92 ± 0.08 [c,d] | 8.98 ± 0.63 [b] | 9.41 ± 0.66 [a,b] |
| CP-20 | 89.53 ± 0.29 [c] | −2.4 ± 0.09 [b,c] | 10.61 ± 1.01 [c] | 11.00 ± 1.03 [b,c] |
| CP-30 | 88.61 ± 0.47 [b] | −2.23 ± 0.25 [b] | 12.43 ± 2.92 [d] | 13.07 ± 2.79 [c] |
| CP-40 | 87.83 ± 0.47 [a] | −2.70 ± 0.42 [a] | 16.10 ± 1.33 [e] | 16.72 ± 1.40 [d] |

Values are given as mean ± standard deviation. Different letters in the same column indicate significantly different ($p < 0.05$) when analyzed by Duncan's New Multiple Range Test.

### 3.4. Mechanical Properties

Mechanical properties are useful parameters for evaluating the food packaging materials. As shown in Table 3, with the increase of corn peptide concentration, the tensile strength of films increased but then decreased. The highest tensile strength was found in CP-30, the possible reason was that the covalent interactions between CMCS and CP in the Maillard reaction which resulted in an increase in network microstructure and intermolecular forces, thereby increasing the tensile strength [35]. The tensile strength of CP-40 was lower (18.73%) than CP-30 due to the decrease of Maillard reaction progression, and, it also could be proven from the degree of glycosylation (Figure 1). With the increase of corn peptide concentration, the elongation of films decreased due to the increase of the number of hydrogen bonds between the carboxyl group in CMCS and the hydroxyl group in CP [36].

**Table 3.** Tensile strength, elongation of films.

| Films | Tensile strength (M pa) | Elongation (%) |
|---|---|---|
| Control | 19.80 ± 0.22 [a,b] | 19.88 ± 0.36 [c,d] |
| CP-10 | 20.10 ± 0.42 [a,b] | 18.13 ± 0.30 [c] |
| CP-20 | 21.33 ± 0.33 [b] | 18.13 ± 0.52 [c] |
| CP-30 | 23.87 ± 0.12 [c] | 16.75 ± 0.28 [b] |
| CP-40 | 19.40 ± 1.71 [a] | 14.55 ± 0.94 [a] |

Values are given as mean ± standard deviation. Different letters in the same column indicate significantly different ($p < 0.05$) when analyzed by Duncan's New Multiple Range Test.

### 3.5. Water Vapor Permeability (WVP)

The transfer of ambient atmospheric vapor to packaged food affects the shelf life of food. Water vapor permeability values of the films are shown in Figure 2. The WVP of Control film was 24.97 ± 0.11 (g mm m$^{-2}$ day$^{-1}$ kpa), which was closely related to the hydrophilic property of CMCS [15]. With the increase of corn peptide concentration, the WVP of films decreased. WVP of the films are believed to depend on the number of available polar groups in polymer [14]. In this work, hydrogen bonds could form between COO- groups of CMCS and -NH$_2$ groups of the CP, reducing the number of available polar groups. The lowest WVP (18.27 ± 0.48 g mm m$^{-2}$ day$^{-1}$ kpa) was found in CP-40, which could be a consequence of an increase in viscosity caused by adding excessive concentration of CP.

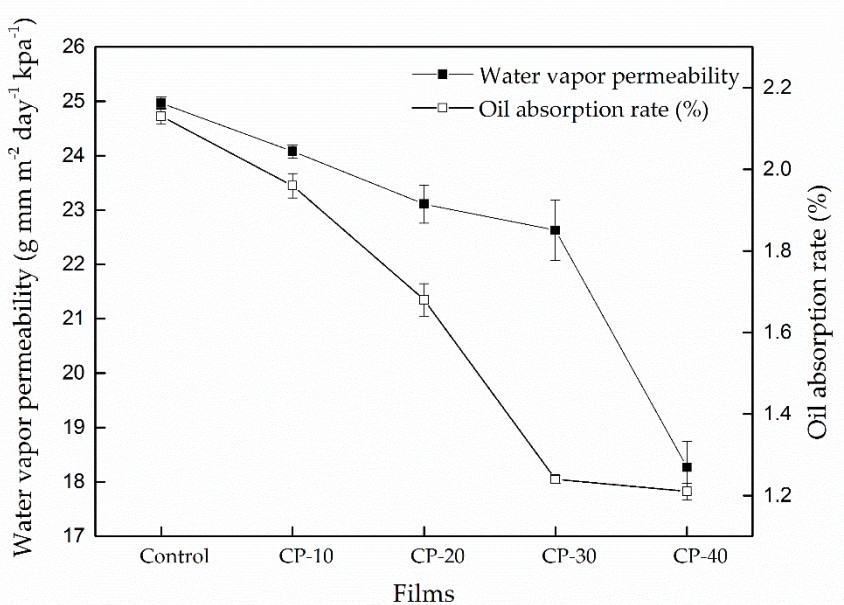

**Figure 2.** Water vapor permeability and oil absorption rate of films.

### 3.6. Oil Resistance

In this experiment, the oil resistance of films was characterized by the oil absorption rate (OAR). It can be seen from Figure 2, the OAR of films incorporated CP was 7.98–43.19% lower than that of Control film (without CP). One possible reason was that the Maillard reaction product based on CMCS and CP filled the structural gaps in the CMCS polysaccharide chain, reduced the interstitial space in the CMCS matrix, and prevented oil molecules from passing through the films. The lower the OAR value, the higher the oil resistance [30]. This trend in the OAR value of films was favorable. Therefore, Maillard reaction product based on CMCS and CP could improve the oil resistance of films and prevent oil transfer and diffusion. It is a good practical value in packaged food.

### 3.7. Antioxidant Activity

Antioxidant packaging is one of the main categories of active packaging to extend the shelf life of food. The Control film (without CP) was shown the weak antioxidant activity for DPPH radical scavenging (Table 4). With the increase of corn peptide concentration, the DPPH radical scavenging of films significantly ($p < 0.05$) increased but then decreased. The DPPH radical scavenging activity of films could be related to the decrease in the number of $-NH_2$ groups, because more active amino groups could donate more hydrogen to react with DPPH radical [37]. The highest DPPH radical scavenging was found in CP-30, more than 13 folds in comparison with the Control (without CP), due to the formation of highly antioxidant compounds (melanoidins), which was supported the results of the degree of glycosylation and browning intensity (Abs 420 nm) (Figure 1). The DPPH radical scavenging of CP-40 was lower (13.81%) than CP-30. The possible reason was that the viscosity of the system was increased. Therefore, the increase in viscosity also limits the migration of molecules and the progress of reactions.

The capacity of films to convert $Fe^{3+}$ into $Fe^{2+}$ was researched. From Table 4, the Control showed the lowest activity. With the increase of corn peptide concentration, the reducing power of films increased but then decreased. The highest reducing power was found in CP-30, more than 2 folds in comparison with the Control (without CP), due to the formation of new antioxidant compounds (melanoidins) during the Maillard reaction between CP and CMCS, which could donate the electron and hydrogen atom to react with free radicals and terminate the free radical chain reactions [17,37]. The reducing power of CP-40 was lower (24.32%) than CP-30. The possible reason was that the viscosity of the system was increased.

**Table 4.** DPPH free radical scavenging activity, reducing power, total antioxidant activity, and antibacterial activities of films.

| Films | DPPH Free Radical Scavenging Activity (%) | Reducing Power (OD$_{700 \text{ nm}}$) | Total Antioxidant Activity ($\alpha$-Tocopherol $\mu$mol/mL) | Diameter of Inhibition Zone (mm) | |
|---|---|---|---|---|---|
| | | | | *E. Coli* | *S. Aureus* |
| Control | 1.02 ± 0.31 [a] | 0.2470 ± 0.01 [a] | 21.47 ± 0.27 [a] | 6.23 ± 0.12 [a] | - |
| CP-10 | 7.89 ± 1.52 [b] | 0.3850 ± 0.02 [b] | 22.21 ± 0.12 [b] | 6.56 ± 0.46 [b] | - |
| CP-20 | 11.92 ± 2.05 [c] | 0.3956 ± 0.01 [c] | 24.69 ± 0.26 [c] | 6.66 ± 0.31 [b,c] | - |
| CP-30 | 14.27 ± 1.42 [d] | 0.5880 ± 0.07 [e] | 32.69 ± 0.56 [d] | 6.88 ± 0.27 [c] | - |
| CP-40 | 12.30 ± 1.01 [c] | 0.4450 ± 0.03 [d] | 28.28 ± 0.38 [e] | 6.76 ± 0.88 [b,c] | - |

Values are given as mean ± standard deviation. Different letters in the same column indicate significantly different ($p < 0.05$) when analyzed by Duncan's New Multiple Range Test.

Table 4 showed the total antioxidant activity of films determined from regression equation of the calibration curve (y = 0.011C + 0.0049; r$^2$ = 0.987) and the change of the total antioxidant activity with the concentration of CP. The Control showed the lowest total antioxidant activity. With the increase of corn peptide concentration, the total antioxidant activity of films significantly ($p < 0.05$) increased but then decreased. The highest total antioxidant activity was found in CP-30, the possible reason was that the combinations of CP and CMCS forms Maillard reaction products, which could react with Mo (VI) to convert it to more stable molecules, Mo (V), by donating electrons [32].

### 3.8. Antibacterial Activities

Microbial activity is one of the main causes of food spoilage, so it is necessary to evaluate the antibacterial activity of food active packaging. Table 4 shows the antibacterial activity of films exhibited against *S. aureus* and *E. coli* by the filter disc inhibition zone. As shown in Table 4, it was obvious that the concentration of CP improved the antibacterial activity of films against *E. coli*. With the increasing of CP concentration, the antibacterial activity of films was increased. Compared with the Control film, the diameter of inhibitory zone of films extract solution with 10%, 20%, 30%, and 40% concentration of CP against *E. coli* increased 5.3%, 6.9%, 10.43%, and 8.5%, respectively. Thus, it can be seen the film extract solution with 30% CP exhibited the best antibacterial activity. This is owing to the antibacterial compounds (mainly melanoidins) and the by-product H$_2$O$_2$ formed during the Maillard reaction between CMCS and CP [26,38,39]. However, there were no significant difference ($p > 0.05$) among CP-10, CP-20, and CP-40. Since Gram-negative bacteria (*E. coli*) have a protective cell film on the surface, the melanoidins was not particularly easy to inhibit *E. coli* [40]. The antibacterial activity of CP-40 was lower (13.64%) than CP-30. The possible reason was owing to the degree of glycosylation of films (Figure 1).

Regarding the antibacterial activity of films against *S. aureus*, the inhibitory zone of films extract solution with 10%, 20%, 30%, and 40% of CP was not observed. The possible reason is that the cell film of *S. aureus* is thick.

### 3.9. Attenuated Total Reflectance-Fourier Transform Infrared Analysis (ATR-FTIR)

ATR-FTIR was performed to study the intermolecular interaction between CMCS and CP. Figure 3 shows the spectra of films in the range of 4000–500 cm$^{-1}$. In general, the ATR-FTIR spectrum of the films showed the same main peak, but the amplitude peak was different, depending on the CP concentration. In the Control film spectrum, the absorption band at 3285.02 cm$^{-1}$ corresponded to the -NH$_2$ and -OH groups of CMCS [41]; and the peaks at 1581.59 cm$^{-1}$ and 1408.10 cm$^{-1}$ were assigned to antisymmetric and symmetric vibrations for the COO- group, respectively [42]. After incorporation of CP, the vibrations for the COO- peaks of the Control, which were located at 1581.59 cm$^{-1}$ and 1408.10 cm$^{-1}$, respectively, and shifted to 1579.66 cm$^{-1}$ and 1406.74 cm$^{-1}$ for CP-40. In addition, the intensity of absorption in these COO- bands decreased with increasing the concentration of CP. The vibrations for the -NH$_2$ peaks of the Control, which were

located at 3285.02 cm$^{-1}$ shifted to 3281.25 cm$^{-1}$ for CP-40, the intensity of absorption decreased with increasing the concentration of CP. This result was indicative of interactions between the -NH$_2$, -OH groups of CP, and the COO- groups of CMCS. Moreover, a new peak appeared in films incorporated CP at 2849.52.0 cm$^{-1}$ and 2955.61 cm$^{-1}$, which could be caused by the Maillard reaction.

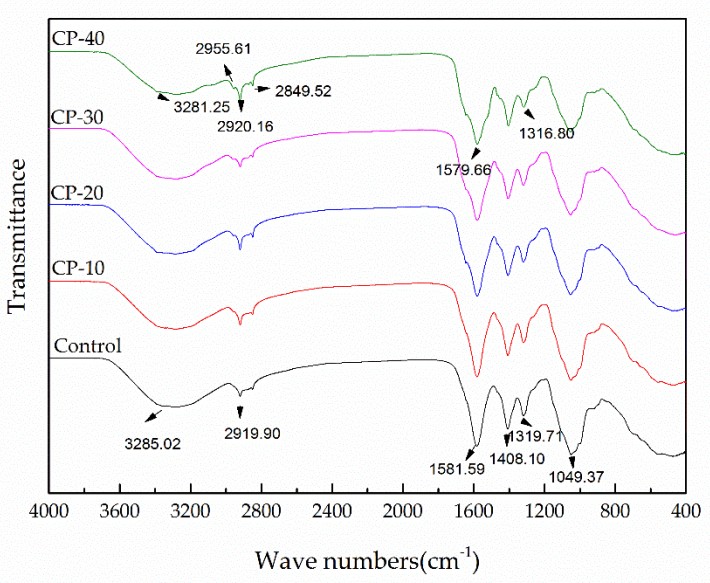

**Figure 3.** ATR-FTIR spectrum of films.

## 4. Conclusions

The incorporation of CP significantly affected the physicochemical properties and biological activities of films. Among the five films, CP-30 possessed the highest degree of glycosylation, tensile strength, antioxidant, and antibacterial activities. ATR-FTIR was confirmed the intermolecular interactions between CP and CMCS. Therefore, the Maillard reaction products of the films based on CMCS and CP would be used as an antioxidant and antibacterial packaging system. Nevertheless, further research is needed, such as analyzing the shelf life of food.

**Author Contributions:** Preparation of films and writing of the manuscript, L.W.; characterization of films and data analysis, L.L.; drawing and discussion, K.W.; interpretation of data and editing, Y.F.; review and approval for submission of the manuscript, H.H.; All authors have read and agreed to the published version of the manuscript.

**Funding:** This research was supported by Opening Project of the Key Laboratory of Bionic Engineering (Ministry of Education), Jilin University (KF20200006).

**Institutional Review Board Statement:** Not applicable.

**Informed Consent Statement:** Not applicable.

**Data Availability Statement:** Data are contained within the current manuscript.

**Conflicts of Interest:** The authors declare no conflict of interest.

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
