# Peer review of "Physicochemical Properties and Biological Activity of Active Films Based on Corn Peptide Incorporated Carboxymethyl Chitosan"

_coatings, doi:10.3390/coatings11050604_

Round 1

Reviewer 1 Report

The authors have developed active films based on carboxymethyl chitosan incorporated corn peptide.  The authors have studied different properties of the films, including thickness, opacity, moisture content, color, mechanical properties, water vapor permeability, and oil resistance. But, this study seems incomplete because the important surface property of the film has not been discussed so far. The readership will benefit if the surface chemistry data is included in the manuscript.  I ask the authors to study the surface property like the homogeneity of the surface, stability of the film (how long). Some surface chemistry like the Langmuir monolayer study can be also performed. 

Author Response

Thank you for your comments concerning our manuscript entitled “Physicochemical properties and biological activity of active films based on corn peptide incorporated carboxymethyl chitosan” (coatings-1217455). Those comments are all valuable and very helpful for revising and improving our paper, as well as the important guiding significance to our researches. We have studied comments carefully and have made correction which we hope meet with approval.

Reviewer 2 Report

Comments to Author:

Manuscript Number: coatings-1217455

Title: Physicochemical properties and biological activity of active films based on corn peptide incorporated carboxymethyl chitosan.

Widely used conventional plastics have many advantages, however their resistance to biological agents causes in negative impact on the environment, so the use apply of (bio)degradable polymers should become widespread due to growing interest in sustainability, organic recycling and environmental issues. Publications concerning environmentally friendly active packaging materials is very needed.

On the one hand, I found the paper to be well written and well described. But I found some small inaccuracies. Therefore, I recommend that a minor revision is warranted.

  • Line 70 - After entering the full name for the first time, please use abbreviations through the text. (carboxymethyl chitosan)
  • Line 77 – Word “Constant” is written with an uppercase. Please change that and correct it throughout the text.
  • Equation number 4 - is in error. Description does not match the content.
  • Table 1, 2 and 4 - The p value is uppercase. Please change that.
  • Line 325 – What does mean in this case “not obvious”? You did not observe any inhibitory zone, or it was negligible.
  • Line 347-348 - According to figure 2 the CP-30 sample did not achieve the lowest value of oil resistance.

Author Response

(The authors gave the same response as above.)

Reviewer 3 Report

The manuscript entitled "Physicochemical properties and biological activity of active 2 films based on corn peptide incorporated carboxymethyl 3 chitosan" is a very interesting and well structured one. After reading the additionally sent Tables and Figures, I consider that the manuscript may be published after a minor revision:
The abstract lacks a clear definition of the purpose of this paper (I suggest a reformulation as detailed as possible)
In the introduction I suggest adding information related to the use of corn pectin in making biofilms.
I suggest unitary reporting of results (Table 1, Thickness - reporting of results to 2 decimal places)
Bibliographic references do not comply with the requirements of the journal

Author Response

(The authors gave the same response as above.)

Reviewer 4 Report

The uploaded manuscript is incomplete- figures, tables and supporting data are missing. Please upload including all this information.

Author Response

(The authors gave the same response as above.)

Round 2

Reviewer 1 Report

The authors have not addressed my concern. The manuscript is incomplete with the surface chemistry study of the film that they have developed. 

Author Response

Thank you for your letter and for the reviewers’ comments concerning our manuscript entitled “Physicochemical properties and biological activity of active films based on corn peptide incorporated carboxymethyl chitosan” (coatings-1217455). Those comments are all valuable and very helpful for revising and improving our paper, as well as the important guiding significance to our researches. We have studied comments carefully and have made correction which we hope meet with approval.

Reviewer 4 Report

Accept

Author Response

 Thank you very much for your comments and suggestions concerning our manuscript entitled “Physicochemical properties and biological activity of active films based on corn peptide incorporated carboxymethyl chitosan” (coatings-1217455), and hope you happy every day.